# Effects of Resistance Training on Spasticity in People with Stroke: A Systematic Review

**DOI:** 10.3390/brainsci14010057

**Published:** 2024-01-06

**Authors:** Juan Carlos Chacon-Barba, Jose A. Moral-Munoz, Amaranta De Miguel-Rubio, David Lucena-Anton

**Affiliations:** 1Department of Nursing and Physiotherapy, University of Cádiz, 11009 Cadiz, Spain; juancarlos.chaconbarba@alum.uca.es (J.C.C.-B.); david.lucena@uca.es (D.L.-A.); 2Biomedical Research and Innovation Institute of Cadiz (INiBICA), 11009 Cadiz, Spain; 3Department of Nursing, Pharmacology and Physiotherapy, University of Cordoba, 14004 Cordoba, Spain; z42mirua@uco.es

**Keywords:** stroke, muscle spasticity, resistance training

## Abstract

Resistance training induces neuromuscular adaptations and its impact on spasticity remains inadequately researched. This systematic review (PROSPERO: CRD42022322164) aimed to analyze the effects of resistance training, compared with no treatment, conventional therapy, or other therapies, in people with stroke-related spasticity. A comprehensive search was conducted up to October 2023 in PubMed, PEDro, Cochrane, Web of Science, and Scopus databases. Selection criteria were randomized controlled trials involving participants with stroke-related spasticity intervened with resistance training. The PEDro scale was used to evaluate the methodological quality. From a total of 274 articles, 23 full-text articles were assessed for eligibility and nine articles were included in the systematic review, involving 225 participants (155 males, 70 females; mean age: 59.4 years). Benefits were found to spasticity after resistance training. Furthermore, studies measuring spasticity also reported benefits to function, strength, gait, and balance. In conclusion, resistance training was superior to, or at least equal to, conventional therapy, other therapies, or no intervention for improving spasticity, as well as function, strength, gait, and balance. However, the results should be taken with caution because of the heterogeneity of the protocols used. Further research is needed to explore the effects of resistance training programs on people with stroke.

## 1. Introduction

Stroke is defined as a sudden loss of neurological function resulting from an infarction or hemorrhage in the brain, spinal cord, or retina and this loss is persistent for over 24 h [1]. It is the second leading cause of death worldwide [2]. In addition, it leads to motor dysfunction and limitations in activities of daily living and quality of life [3]. Spasticity is a motor disorder characterized by an increase in the muscle stretch reflex, accompanied by hypertonia and hyperreflexia, associated with an injury to the upper motor neurons [4]. The neurophysiology of spasticity in stroke involves damage to specific brain areas, such as the superior corona radiata, posterior limb of the internal capsule, thalamus, putamen, premotor cortex, and insula [5]. These brain lesions disrupt the normal inhibitory signals from the brain to the muscles, leading to hyperexcitability of the stretch reflex and increased muscle tone [6]. It is estimated that between 38% and 40% of people with stroke will have some spasticity, with treatment being necessary in 16% of cases. This estimated prevalence varies depending on the time elapsed after the stroke, being 27% in the first month, and 42.6% in periods longer than 3 months [7].

Voluntary muscle contraction and recruitment of muscle fibers in people with stroke may be highly complex because of the exorbitant response it provokes immediately, so it is usually avoided [8]. Traditional stroke treatment programs excluded muscle strengthening because it overexcited the muscle tracts, increasing the spastic process; while muscle weakness was considered as a secondary factor in limiting motor function [9,10]. Nonetheless, it is known that paretic muscle atrophy strongly correlates with reduced fitness levels, so resistance training has the potential to support normal muscle functioning within the affected limb and may counteract the stroke-related decrease of physical fitness as well as the stroke-related sarcopenia [11,12,13]. Exercise has been shown to create an optimal environment for neuroplasticity in the primary motor cortex and other areas of the brain related to motor control, leading to enhanced motor learning and function [14].

In recent years, it has been shown that there is great evidence of the benefits of resistance training programs in different populations [15,16,17,18,19]. In this sense, resistance training induces a development of power, hypertrophy, and muscle strength, by generating neural and structural adaptations in the medium and long term [20], besides the increase in force production that implies the development of muscle size and cross-section, as well as the modification of the arrangement of muscle fibers [21]. Furthermore, there is an improvement in intermuscular coordination, which is manifested in an intensification in the relaxation capacity of the antagonist muscles during agonist contraction [22], and a greater recruitment of motor units in less time, with an optimization of reflex phenomena. This leads to an increase in the speed and strength of muscle contraction [23]. Nevertheless, it is reasonable to assume that these benefits are present in stroke patients and reduce spasticity, but to date, there is no literature indicating the neurophysiological mechanisms that provoke this phenomenon.

In view of this background, the treatment of spasticity in people with stroke is a main therapeutic goal, but the evidence of using resistance training as an intervention for spasticity is still unclear. This issue was analyzed [15,16,17,18,19] in a systematic review and meta-analysis [11], in which the benefits of resistance training in supporting the recovery of stroke patients were analyzed, and no significant improvements with respect to no intervention or other interventions were found in spasticity. This result was based on only two randomized controlled trials (RCT), highlighting the lack of available evidence on this topic. Therefore, there is no systematic review on the uses of resistance training on spasticity that synthetizes the protocols used and serves as a basis for clinical decision making in stroke rehabilitation.

Therefore, the aim of this systematic review was to analyze the effects of resistance training on spasticity in people with stroke. Furthermore, we aimed to explore the implications on function, strength, gait, and balance in addition to the spasticity. Moreover, the resistance training programs and protocols for the treatment of people with stroke will be analyzed.

## 2. Methodology

The present study is a systematic review reported according to the guidelines established in the PRISMA 2020 statement (Preferred Reporting Items for Systematic Reviews and Meta-Analyses) [24] (Appendix A). In addition, this systematic review was registered in the Prospective Register of Systematic Reviews (PROSPERO), register number: CRD42022322164.

### 2.1. Search Strategy and Selection Process

The literature search for this review was conducted up to October 2023, using the following databases: PubMed, PEDro (Physiotherapy Evidence Database), WoS (Web of Science), Scopus, and CENTRAL (Cochrane Controlled Register of Trials). The search strategy was performed through the combination of different keywords and Boolean operators “AND” and “OR”, as shown in Table 1. In the PubMed, CENTRAL, WoS, and Scopus databases, study filters were applied showing only RCT in the case of PubMed, trials in the case of CENTRAL, and articles in the case of WoS and Scopus. For the PEDro database, an advanced search was performed filtering as therapy “strength training”, subdiscipline “neurology”, and methods “clinical trial”. No filters were applied to the date of publication, and no language restriction was established.

The bibliographic information of the retrieved articles was imported into the Mendeley Desktop (version 1.19.4) [25]. An initial manual check was performed to ensure accuracy, followed by grouping and sorting by title to eliminate duplicates. Titles and abstracts were then assessed, and those without human subjects and non-RCTs were discarded. Finally, compliance with inclusion criteria was annotated using the notes tool in Mendeley Desktop [25]. Articles that did not meet the established selection criteria were excluded by evaluating the full-text of the screened articles. The remaining studies were eligible for inclusion in the systematic review.

Two authors (J.C.C.-B. and D.L.-A.) were responsible for the literature search and retrieval of potentially relevant studies. A third reviewer (J.A.M.-M.) took part to reach a consensus when necessary.

### 2.2. Selection Criteria

The criteria defined for the inclusion of this study were based on the PICOS (Patient, Intervention, Outcomes, and Study type) research model [26]: (P) adults with stroke; (I) resistance training isolated or mixed (such as functional exercise, aerobic training or task-oriented training) programs aiming to develop muscle strength; (C) no treatment, conventional therapy, or other therapies; (O) spasticity; (S) RCT. Studies involving non-active interventions for muscle strength improvement were excluded (e.g., electrostimulation). We also excluded trials that had a mix of people with stroke and other populations and did not report outcomes for people with stroke separately.

### 2.3. Data Extraction

The extracted data included the characteristics of the study participants, the duration and sessions performed in each intervention, the characteristics of the intervention, the time and tools of measurement, and data on the results. Data extraction was performed by two independent reviewers (J.C.C.-B. and D.L.-A.). A third reviewer (J.A.M.-M.) participated in resolving conflicts during the process.

### 2.4. Methodological Quality Assessment

The articles included in this review were evaluated using the PEDro scale to assess their methodological quality [27]. The PEDro scale allows users to determine in a simple way the external validity (criterion 1), the internal validity (criteria 2–9), and the statistical information for the interpretation of the results (criteria 10 and 11), being a very useful and specific instrument to evaluate the quality of clinical trials. According to the score obtained on the PEDro Scale, the studies have been classified as low quality (score less than 4), moderate (score of 4–5), good (score of 6–8) or excellent (score of 9–10), with criterion 1 being excluded from the final score [28].

The assessment was performed independently by two authors (J.C.C.-B. and J.A.M.-M.). A third reviewer (A.D.M.-R.) participated to establish a consensus when necessary.

## 3. Results

A total of 274 articles were found in a first search and a total of 80 duplicate records were removed. The titles and abstracts of the remaining records (194) were screened and 171 were then excluded due to different reasons (not topic and not RCT). The full-texts of the 23 remaining studies were assessed to verify the compliance of the eligibility criteria. Finally, nine articles were included in this review. The PRISMA flowchart in Figure 1 shows the selection of studies for this systematic review.

### 3.1. Methodological Quality

As shown in Table 2, the mean score of the PEDro scale of the articles included in this review is 5, classified as a moderate mean methodological quality. The studies that score highest on this scale were those of Coroian et al. [29], Dehno et al. [30], and Patten et al. [31], considering them to have a good methodological quality, with a score of 7/10. On the other hand, the study included in this review with the lowest score on the PEDro scale is that of Fernandes et al. [32], with a score of 2/10, therefore having a low methodological quality.

### 3.2. Participants

The characteristics of the participants included in this study are shown in Table 3. A total of 225 participants were included in this review, with the study by Lattouf et al. [37] reporting the largest number (*n* = 37), while the RCT by Fernandes et al. [32] had the smallest (*n* = 16). The mean age was 59.4 years, with Patten et al. [31] reporting the oldest, 72.9 years, and Akbari et al. [33] reporting the youngest, 48.8 years. The number of women included represented 32% of the total number of participants and only Fernandes et al. is gender-balanced [30]. In contrast, the study by Fernandes et al. [32] only presents male subjects.

The median time elapsed after the cerebrovascular event in the included participants was 21.3 months, with only one study presenting subjects with an acute cerebrovascular event [32]. The study by Fernandez et al. [34] had the participants with the longest time elapsed after stroke.

The proportion of patients with an ischemic-hemorrhagic stroke was reported in all of the studies, with the exception of one study [33]. In this way, ischemic stroke occurred in 73.30% of cases.

### 3.3. Interventions

The characteristics of the interventions included in this review are shown in Table 4. Many of the included studies lasted 4 weeks [30,31,33,37], and two extend up to 12 weeks [32,34]. In terms of the number of sessions, three of the articles covered a total of 12 sessions [30,31,33], while the study by Fernandes et al. [32] stood out for having the highest number of sessions, with a total of 48.

Regarding weekly frequency, it was observed that three of the studies implemented 3 sessions per week [29,30,33], and two had the maximum frequency, with 5 weekly sessions each [36,37]. Two articles showed the lowest frequency, with only two sessions per week [34,35]. It should be noted that Coroian et al. [29] and Lattouf et al. [37] performed two daily sessions with a frequency of three days per week and five days per week, respectively.

In relation to the duration of the sessions, considerable variability was observed among the articles. The average duration of the sessions was 70 min, ranging from the shortest session of 30 min [29,36,37] up to the longest of 3 h [33]. It should be considered that two of the three articles that held 30-min sessions performed two sessions a day [29,37].

In terms of the interventions, four studies focused only on resistance training as an exercise modality [29,34,35,36]. Two studies combined resistance training with conventional therapy [30,37], while two others did so with task-oriented training [31,32]. Only one intervention complemented resistance training with aerobic and functional exercises [33]. Six workouts were specifically aimed at the lower limbs [32,33,34,35,36,37], while three focused on the upper limbs [29,30,31], addressing specific aspects, such as wrist, elbow, and wrist, as well as shoulder and elbow.

Regarding the systems used to perform resistance training, three studies used dynamometers [29,30,31], and the remaining studies used a leg press [37], sliding stander [36], knee exercise machine [35], and an inertial flywheel [34]. The remaining two studies did not specify the system used. These findings reflect the heterogeneity in the methods and equipment used in the studies reviewed, although there is a consensus that the exercises employ closed kinetic chains.

The results of the systematic review showed significant variability in exercise intensity, although a considerable number of studies used a load of 70% of repetition maximum, evidencing a common preference for this intensity [29,33,36]. Overall, articles reported a load ranging from 40% to 80% of the maximum repetition. A prominent approach was the inclusion of an inertial exercise, whereby the intensity of the exercise depended on the force applied by the participant himself [34].

Regarding the number of sets and repetitions, the results suggest that, although we did not identify a universally predominant protocol, some notable patterns and trends were evidenced. In terms of the number of repetitions, it was noted that most studies opted for protocols with repetitions ranging from 6 to 15. In particular, the most common protocol consisted of 3 sets of 10 repetitions, this being the standard adopted in several studies [31,34]. Regarding the number of series, a heterogeneous distribution was found. The general preference was to perform between 3 and 5 sets per training session [30,31,32,34,36,37]. However, some studies presented less conventional approaches, such as that of Flansbjer et al. [35], which proposed 2 sets of 6–8 repetitions. As for the speed of execution, it is only specified in three articles, and there is a discrepancy in this aspect [29,30,35].

It should be noted that there seems to be an interest in the combination of the different forms of contraction, looking for concentric, isometric, and concentric phases during the execution of the repetition [31,32,37]. However, even though many do not specify the type of contraction requested, it can be inferred from the type of exercise to be carried out that the concentric contraction stands out.

### 3.4. Outcomes Measures

The outcome measures, measuring instruments, and results obtained by the studies are shown in Table 5.

Concerning the spasticity, four measuring instruments were used, but three of them were versions of the Ashworth scale. Most studies employed the Modified Ashworth Scale [29,32,33,34,35,37]. The remaining studies used the Ashworth Scale [31], the Modified Ashworth Scale [30], and the Biodex system [36], which evaluates the resistance to ankle mobilization at different speeds. The section evaluated coincided with the musculature involved in the resistance exercise. Four of the included studies found a significant improvement between pre and post intervention in EG [30,33,35,36]. Three articles found an improvement in EG compared to CG [30,33,36], while the other studies found no differences between groups. The study by Mun et al. [36] found improvements in all angular velocities evaluated. The remaining five articles found no significant differences pre and post intervention in EG.

The function was evaluated in three articles, all of which used the Upper Limb Fugl-Meyer Motor Assessment (UL-FMMA) [29,30,31]. The study by Patten et al. [31] also assessed the function with the Wolf Motor Function Test-Functional Abilities Scale (WMFT-FAS) and the Functional Independence Measure (FIM). The studies by Dehno et al. [30] and Patten et al. [31] found significant improvements in the EG compared with the CG for improving function, which were also maintained at follow-up time points. Specifically, Dehno et al. [30] found improvements for the UL-FMA score, and Patten et al. [31] for the WMFT-FAS and FIM. The study by Coroian et al. [29] only found pre-post improvements in the EG, but these were not significant in comparison with the CG.

Strength was the most evaluated parameter after spasticity [29,30,33,34,35,37], studying the maximum strength of the regions worked during the intervention. Dynamic force was studied by two of the studies [34,35]. The study by Dehno et al. [30] studied the strength of the paretic side using the Medical Research Council (MRC) scale. All studies found significant improvements in strength after the EG intervention, as well as significant differences between groups, except the study by Coroian et al. [29]. The study by Fernandez et al. [34] found improvements in dynamic strength, but the improvement in isometric strength did not become significant.

For gait evaluation, the Timed “Up & Go” (TUG) was mostly used [34,35,36]. The study by Flansbjer et al. [35] also evaluated gait using the Fast Gait Speed (FGS), and the 6-Minute Walk Test (6MWT). Lattouf et al. [37] assessed gait using the 10-m Walk Test and 6-Minute Walk Test (6MWT). The study by Fernandez et al. [34] found significant improvements in EG compared to CG. The study by Lattouf et al. [37] found improvements between groups in the 6MWT but did not find differences in the 10-m Walk Test. Flansbjer et al. [35] did not find changes after the intervention, but it found improvements in EG compared to CG in the follow-up period in TUG. All studies found improvements after the intervention in EG. Moreover, the improvements were maintained in the follow-up evaluation in the study by Flansbjer et al. [35].

Balance was assessed in three of the articles included in this review using the Berg Balance Scale (BBS), the common tool used [32,34,36]. The study by Mun et al. [36] also used a platform that calculates the load distribution when standing, using it both with open and closed eyes. All studies found a significant improvement after the intervention for both groups.

## 4. Discussion

The purpose of this systematic review was to analyze the current scientific evidence on the use of resistance training programs as a therapeutic option for patients with stroke, and to analyze their effects on spasticity. A total of nine studies studying the application of resistance training programs in stroke populations have been included and reviewed. Benefits to spasticity were reported, as well as to function, strength, gait, and balance, being superior to, or at least equal to, those obtained by the comparison groups. Thus, resistance training programs can bring benefits to people with stroke without causing an increase in spasticity [9,10]. This finding, together with the effectiveness that seems to occur in different parameters related to motor function, makes resistance training an adequate alternative intervention for stroke patients.

Concerning the characteristics of the studies included, samples were relatively small, involving an average of 30 subjects per study. In this sense, it is a common limitation in stroke rehabilitation trials, since it is difficult to obtain large sample sizes because patients are usually treated only in a neurological institution or center, costs are usually high, and inclusion criteria are very narrow [38,39]. Furthermore, profiles of the subjects included had great variability in terms of the time elapsed after stroke, with only one study [32] including patients in an acute stage, and they did not find differences in spasticity, although they did find significant improvements in balance. It is known that most recovery is reached around the third month, with the fastest level of recovery occurring in the first month and a half, because it is the period where the greatest endogenous neuroplasticity occurs [40]. Moreover, the incidence of spasticity in stroke occurs mainly in periods longer than 3 months after stroke, so it may be interesting to know the long-term effects of this intervention in subjects with acute stages of stroke, acting then as a preventive action [7].

Two of the three studies that found improvements in spasticity in EG compared to CG carried out protocols combined with another type of treatment [30,33,36]. Therefore, there was no solid evidence to show that resistance training in isolation improves the degree of spasticity. In this way, Coroian et al. [29] was the only one that performed an isolated resistance training intervention and did not find significant improvements in spasticity. In this regard, isolated resistance training may not be entirely adequate for patient improvement after stroke. According to the current literature [41], combined strength, aerobic, and other physical capacity training would be an appropriate approach for the recovery of stroke patients without increased tone or spasticity.

Regarding the intervention protocol, the superiority of one approach compared to the others is not perceived. However, it seems that there is a consensus to perform between three and five sets, with positive effects in the studies that adopted this number of series in their interventions. Concerning the number of repetitions per set, we found too wide a range to draw conclusions. In this line, the study that opted for the highest number of sets, which in turn contained the highest number of total repetitions, was the one that showed the fewest positive effects [29], which may infer that excessive load on resistance training does not result in beneficial effects in stroke patients. Nevertheless, further studies are needed to confirm this issue.

Despite the variability of the tools used for resistance training, there was agreement on the use of equipment that uses closed kinetic chains. This may be because it offers greater sensory information, which is very favorable in this patient profile [42,43]. There was also some interest in combining different forms of contraction during the execution of the exercise, possibly to take advantage of the different effects they cause [44,45].

In view of the above, the present systematic review provides a comprehensive insight into the effects of resistance training on spasticity among people with stroke. However, several limitations need to be considered. The results should be taken with caution due to the heterogeneity of the participant characteristics and intervention protocols. In this sense, a meta-analysis was not performed because of the high heterogeneity in terms of study interventions and outcome measures, as well as the different body regions assessed, so a meta-analysis is not congruent enough to extract a quantitative synthesis that adds qualitative value to the results of the studies analyzed. Also, the number of articles with long-term follow-up was small, and evidence on the long-term effects of resistance training programs is limited. The inclusion criteria, while well-defined using the PICOS model, might have introduced bias due to the exclusion of certain types of physical interventions for muscle strength improvement. In addition, the lack of information about the specific duration of resistance training in multimodal programs was not reported by some studies and therefore no solid conclusions could be drawn on this issue.

Despite these limitations, the systematic review aims to contribute significantly to understanding the relationship between resistance training and spasticity post-stroke. The comprehensive evidence from this review enables clinicians to make informed decisions when considering the inclusion of resistance training in post-stroke rehabilitation plans. Recognizing its potential benefits in reducing spasticity, as well as improving critical functional areas such as strength, gait, and balance, enables the development of personalized and thorough treatment strategies. Furthermore, by highlighting existing gaps and limitations, this review acts as a catalyst for future research efforts. Finally, this systematic review offers clinicians valuable information for refining rehabilitation strategies, which may result in improved functional performance and quality of life for post-stroke, spastic patients.

## 5. Conclusions

Resistance training programs were superior to, or at least similar to, no intervention, conventional therapy, or other therapies for managing spasticity. Furthermore, other additional benefits were found to function, strength, gait, and balance in people with stroke. Therefore, the inclusion of this therapy in clinical practice could have a positive impact on people with stroke. However, there was no solid consensus on the optimal training protocol. It seems that the use of closed kinetic chains and the performance of various forms of contraction are those reporting the best results. Despite the results obtained, they should be taken with caution due to the heterogeneity in terms of participants and intervention protocols. We encourage authors to conduct well-designed research protocols including follow-up assessments to explore the long-term effects of resistance training in people with stroke.

## Figures and Tables

**Figure 1 brainsci-14-00057-f001:**
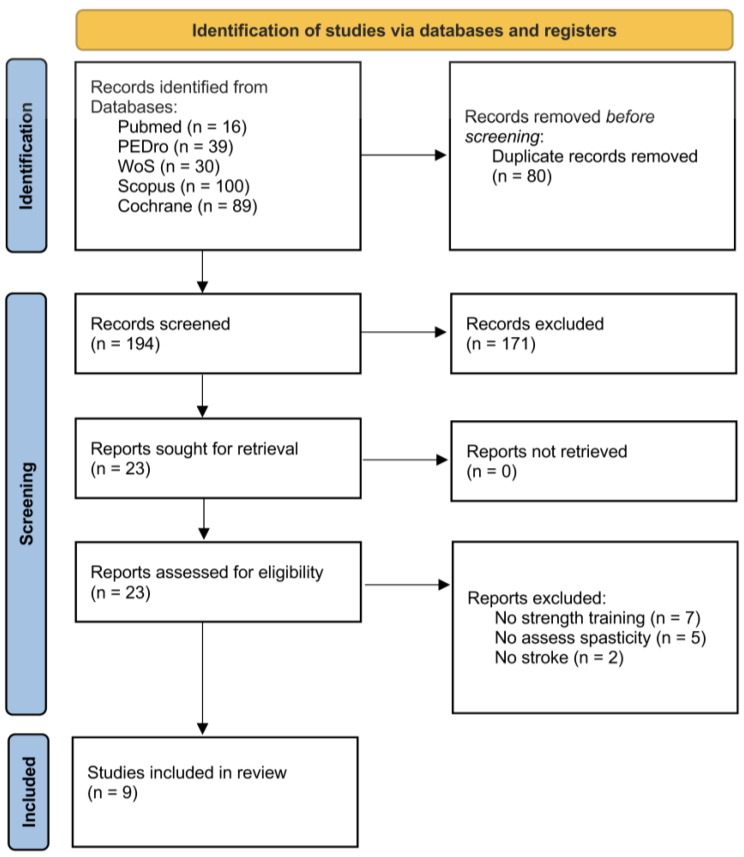
PRISMA flowchart showing the algorithm for the selection of eligible studies.

**Table 1 brainsci-14-00057-t001:** Search strategy.

Database	Number of Articles	Search Strategy
PubMed	16	(“stroke” OR “cerebrovascular accident” OR “hemiparesis” OR “hemiplegia”) AND (“Resistance Training” OR “Strength Training” OR “strengthening”) AND (“Muscle Spasticity” OR “Muscle Hypertonia” OR “Spastic*” OR “Muscle Tonus”) Filter applied: Randomized Controlled Trial
PEDro	39	(“Stroke” AND “spasticity) Therapy: strength training Subdiscipline: neurology Methods: clinical trial
Web of Science (All Databases)	30	TS = ((“stroke” OR “cerebrovascular accident” OR “hemiparesis” OR “hemiplegia”) AND (“Resistance Training” OR “Strength Training” OR “strengthening”) AND (“Muscle Spasticity” OR “Muscle Hypertonia” OR “Spastic*” OR “Muscle Tonus”)) Document Types: Clinical Trial
Scopus	100	TITLE-ABS-KEY((“stroke” OR “cerebrovascular accident” OR “hemiparesis” OR “hemiplegia”) AND (“Resistance Training” OR “Strength Training” OR “strengthening”) AND (“Muscle Spasticity” OR “Muscle Hypertonia” OR “Spastic*” OR “Muscle Tonus”)
CENTRAL	89	(“stroke” OR “cerebrovascular accident” OR “hemiparesis” OR “hemiplegia”) AND (“Resistance Training” OR “Strength Training” OR “strengthening”) AND (“Muscle Spasticity” OR “Muscle Hypertonia” OR “Spastic*” OR “Muscle Tonus”)

**Table 2 brainsci-14-00057-t002:** PEDro scale score for the studies included in this systematic review.

Study	1	2	3	4	5	6	7	8	9	10	11	Total
Akbari et al. (2006) [33]	-	Yes	No	Yes	No	No	Yes	No	No	Yes	Yes	5
Coroian et al. (2018) [29]	-	Yes	No	Yes	No	No	Yes	Yes	Yes	Yes	Yes	7
Dehno et al. (2021) [30]	-	Yes	Yes	Yes	No	No	Yes	Yes	No	Yes	Yes	7
Fernandes et al. (2015) [32]	-	No	No	No	No	No	No	Yes	No	No	Yes	2
Fernandez et al. (2016) [34]	-	Yes	No	Yes	No	No	Yes	Yes	No	Yes	Yes	6
Flansbjer et al. (2008) [35]	-	Yes	No	Yes	No	No	No	Yes	Yes	Yes	No	5
Mun et al. (2019) [36]	-	Yes	No	Yes	No	No	No	No	No	Yes	Yes	4
Lattouf et al. (2021) [37]	-	Yes	No	Yes	No	No	No	No	No	Yes	Yes	4
Patten et al. (2013) [31]	-	Yes	Yes	Yes	No	No	Yes	Yes	No	Yes	Yes	7

Range: 0–10. Item 1 is not used in the method score. Item 1: Eligibility criteria; Item 2: Random allocation; Item 3: Concealed allocation; Item 4: Baseline similarity; Item 5: Subject blinding; Item 6: Therapist blinding; Item 7: Assessor blinding; Item 8: >85% follow-up; Item 9: Intention-to-treat analysis; Item 10: Between-group statistical comparison; and Item 11: Point and variability measures.

**Table 3 brainsci-14-00057-t003:** Main characteristics of the participants included in the systematic review.

Studies	Number of Participants/EG:CG	Age (SD)	Male:Female	Time After Stroke (SD)	Ischemic:Hemorrhagic
Akbari et al. (2006) [33]	N:34 17:17	EG 49.3 (7.1) years	EG 10:7	EG 34.5 (26.37) months	EG ND
		CG 48.8 (3) years	CG 9:8	CG 35.3 (27.5) months	CG ND
Coroian et al. (2018) [29]	N: 20 10:10	EG 63.6 (12.6) years	EG 8:2	EG 32.2 (12.8–629.6) months	EG 9:1
		CG 63.6 (10.6) years	CG 8:2	CG 29.1 (7.6–90.1) months	CG 7:3
Dehno et al. (2021) [30]	N: 26 13:13	EG 53 (9.36) years	EG 7:6	EG 95.22 (37.14) days	EG 11:2
		CG 49.77 (15.48) years	CG 6:7	CG 101.62 (32.39) days	CG 12:1
Fernandes et al. (2015) [32]	N: 16 9:7	EG 58 (6) years	EG 9:0	EG 15 (5) days	EG 6:3
		CG 58 (7) years	CG 7:0	CG 17 (4) days	CG 5:2
Fernandez et al. (2016) [34]	N: 29 14:15	EG 61.2 (9.8) years	EG 11:3	EG 3.5 (3.6) years	EG 9:5
		CG 65.7 (12.7) years	CG 11:4	CG 4.3 (4.9) years	CG 11:4
Flansbjer et al. (2008) [35]	N: 24 15:9	EG 61 (5) years	EG 9:6	EG 18.9 (7.9) months	EG 12:3
		CG 60 (5) years	CG 5:4	CG 20.0 (11.6) months	CG 6:3
Mun et al. (2019) [36]	N:20 10:10	EG 53.1 (13.4) years	EG 8:2	EG 20.3 (14.4) months	EG 3:7
		CG 54.0 (9.1) years	CG 8:2	CG 15.8 (10.2) months	CG 7:3
Lattouf et al. (2021) [37]	N: 37 19:18	EG 65.1 (11.7) years	EG 11:8	EG 11.61 (4.07) months	EG 14:5
		CG 68.7 (12.4) years	CG 11:7	CG 12.26 (5.41) months	CG 14:4
Patten et al. (2013) [31]	N: 19 9:10	GA 64.7 (9.7) years	GA 6:3	GA 14.7 (2.7) months	GA 7:2
		GB 72.9 (11.1) years	GB 9:1	GB 11.4 (4.3) months	GB 7:3

CG, Control group; EG, Experimental group; GA, Group A; GB, Group B; ND, Not described.

**Table 4 brainsci-14-00057-t004:** Main characteristics of the interventions included in the systematic review.

Studies	Duration of Intervention; Frequency of Sessions; Session Time	Intervention
Akbari et al. (2006) [33]	4 weeks; 3 weekly sessions; 3 h per session	EG: 3-part program. Part 1: Standing, walking, and aerobic conditioning exercises. Part 2: Functional exercises. Part 3: Strengthening of the lower limbs, with concentric contraction at 70% 1RM, or synergistic contractions for weakened muscles. Ten repetitions of each exercise for each muscle group
		CG: Same protocol not including the 3rd part
Coroian et al. (2018) [29]	6 weeks; 3 weekly sessions; 2 daily sessions of 30 min	EG: Isokinetic strengthening of the elbow and wrist with dynamometer. A total of 10 min of warm-up (36 reps at 20% 1RM and 15–30° per second) + 30 min of session (six sets of eight reps at 40–70% 1RM, at 15–45° per second)
		CG: 45 min of passive elbow and wrist mobilization with dynamometer
Dehno et al. (2021) [30]	4 weeks; 3 weekly sessions; 60 min per session for CG and 45 min for EG	EG: CT + Unilateral Resistance Training for Wrist Extensors with Isokinetic Dynamometer. Five sets of six concentric repetitions, at 60°/second, with 2-min breaks between sets
		CG: CT
Fernandes et al. (2015) [32]	12 weeks; 4 weekly sessions; 70 min per session	EG: Task-oriented training + Lower limb strengthening affect: three sets of 10 reps, increasing resistance in different positions for each muscle group
		CG: Task-oriented training
Fernandez et al. (2016) [34]	12 weeks; 2 weekly sessions; ND	EG: Unilateral strengthening in the lower limbs with a leg press with an inertial flywheel. Four sets of seven reps maximum, with 3 min of recovery between sets
		CG: Non-intervention
Flansbjer et al. (2008) [35]	10 weeks; 2 weekly sessions; 90 min per session	EG: Lower limb strengthening on knee exercise machine. Warm-up (5 min. of stationary bike, five reps without resistance and five reps at 25% 1RM) + Session (two sets of six to eight reps at 30–40 s per set, at 80% 1RM). After training, passive stretching of the muscles
		CG: Usual daily activities
Mun et al. (2019) [36]	6 weeks; 5 weekly sessions; 30 min per session	EG: Strengthening of the lower limbs in a sliding stander. Warm-up and cool-down (reps for 5 min and 25% 1RM) + session (20 min, 3 sets of 15 to 20 reps at 70% 1RM)
		CG: CT
Lattouf et al. (2021) [37]	4 weeks; 5 weekly sessions; 2 sessions of 30 min	EG: CT + Resistance training in lower limbs in horizontal press (3 phases: concentric, static, and eccentric). Three sets of five repetitions, at 40% 1RM in the first two phases, and at 60% 1RM in the last phase
		CG: CT
Patten et al. (2013) [31]	4 weeks for each intervention, with 4 weeks off between both interventions; 3 weekly sessions for each intervention; 75 min per session	Functional Physical Therapy Intervention: Functional tasks with progression of six objectives and nine activity categories
		Hybrid Intervention: Functional Physical Therapy (20–30 min) + shoulder and elbow resistance training with dynamometer (35 min, 3 sets of 10 reps of shoulder abduction/adduction, shoulder flexion/extension, external/internal rotation of the shoulder, and flexion/extension of the elbow; first set in eccentric and the next two in concentric, with gradual increase in speed)

1RM, One-Repetition Maximum CG, Control Group; CT, Conventional Therapy; EG, Experimental Group; ND, Not Described.

**Table 5 brainsci-14-00057-t005:** Evaluation and results of the articles included in the systematic review.

Studies	Outcome Measures	Results
Akbari et al. (2006) [33]	Two measurement time points (pre and post intervention) - Spasticity: MAS - Strength: Maximum force evaluated with dynamometer.	Spasticity: There was a significant decrease in quadriceps spasticity in EG (*p* < 0.0001), but no change in CG (*p* = 0.055). There was a significant decrease in gastrocnemius spasticity in both EG (*p* < 0.0001) and CG (*p* = 0.041). A significant decrease in spasticity was found in EG compared to CG in both quadriceps (*p* = 0.034) and gastrocnemius (*p* = 0.001).
		Strength: There was an increase in strength in all muscles in EG on both the affected side (*p* < 0.0001) and the non-affected side (*p* < 0.0001). No differences in strength were found in CG, except in hip and knee extensors (*p* < 0.0001) and ankle flexors (*p* = 0.008) on the non-affected side, and hip extensors (*p* = 0.003) and knee extensors (*p* < 0.0001) on the affected side. A significant increase in muscle strength was found in EG compared to CG (*p* < 0.0001), except in knee extensors (*p* = 0.184).
Coroian et al. (2018) [29]	Four measurement time points (pre and post intervention, 3 months after and 6 months after) - Spasticity: MAS. - Function: UL-FMA - Strength: Maximum force evaluated with dynamometer	Spasticity: No significant differences in spasticity were found between the different time points in both groups (*p* = 0.4). No significant differences were found between groups (*p* = 0.98). Function: No significant differences were found pre and post intervention in the total UL-FMA score between the two groups (*p* = 2). No differences were found in the proximal UL-FMA score. In subsequent time points, no significant differences were found in UL-FMA scores between the two groups. There was a significant improvement for EG in the total UL-FMA score pre and post intervention (*p* < 0.01), which was maintained at 3 months (*p* < 0.01) and 6 months (*p* < 0.01) Strength: No significant differences were found in changes in dynamometer scores in different time points for elbow flexors (*p* = 0.2), elbow extensors (*p* = 0.3), wrist flexors (*p* = 0.1) and wrist extensors (*p* = 0.1). No significant differences were found between groups in dynamometer scores for elbow flexors (*p* = 0.2), elbow extensors (*p* = 0.8), wrist flexors (*p* = 0.2) or wrist extensors (*p* = 0.3).
Dehno et al. (2021) [30]	Two intervention measurement time points (pre and post intervention) - Spasticity: MMAS. - Function: UL-FMA. - Strength: On the less affected side, maximum force evaluated with dynamometer. On the most affected side, the Medical Research Council scales.	Spasticity: A significant improvement was found in the MMAS score in EG (*p* = 0.002). There were no differences in the MMAS score in CG (*p* = 0.165). A significant change in spasticity was found in EG compared to CG (*p* = 0.014). Function: Significant improvements were found for both groups between pre and post intervention in the total UL-FMA score (EG *p* = 0.001, CG *p* = 0.001). The improvement in the UL-FMA score in the EG was significantly greater than in the CG (*p* = 0.04). Strength: There was a significant improvement in the strength of the less affected side in EG (*p* = 0.001). There were no significant differences in the strength of the less affected side in CG (*p* = 0.106). There was a significant improvement in the strength of the less affected side between EG and CG (*p* = 0.001). There was a significant improvement in the strength of the most affected side compared to the start of treatment in EG (*p* = 0.001) and CG (*p* = 0.001). There was a significant improvement in EG compared to CG (*p* = 0.029).
Fernandes et al. (2015) [32]	Two measurement time points (pre and post intervention) - Spasticity: MAS. - Balance: BBS	Spasticity: There were no significant differences between or within the groups (*p* ≥ 0.05). Balance: The results in the BBS showed significant differences in both groups (EG *p* = 0.002, CG *p* = 0.008). The comparison between groups after the intervention showed that there was a significant difference, with the EG achieving a greater improvement (*p* = 0.008).
Fernandez et al. (2016) [34]	Two measurement time points (pre and post intervention) - Spasticity: MAS - Strength: Maximum isometric and dynamic force evaluated in leg press - Gait: TUG. - Balance: BBS.	Spasticity: There were no differences between or within the groups. Strength: Differences were found in the isometric strength of the affected leg after the intervention in EG (*p* = 0.02). An improvement in isometric strength was found in EG compared to CG, although it was not significant (*p* = 0.06). There was an improvement in the dynamic strength of both legs after the intervention in EG (*p* = 0.03). A significant improvement in dynamic strength was found in EG compared to CG (*p* = 0.03). Gait: An improvement was found in TUG after the intervention in EG (*p* = 0.01) but not in CG. Significant improvements were found in TUG in EG compared to CG (*p* = 0.04). Balance: An improvement in BBS after the intervention was found in both EG (*p* < 0.001) and CG (*p* = 0.01). Significant improvements were found in EG compared to CG (*p* < 0.001).
Flansbjer et al. (2008) [35]	Three measurement time points (pre and post intervention, and 5 months after the intervention) - Spasticity: MAS. - Strength: Dynamic strength on a knee exercise machine. Maximum isometric force assessed with a dynamometer. - Gait: TUG, Fast Gait Speed, and 6MWT.	Spasticity: A significant improvement in the MAS score was found after the intervention in EG (*p* < 0.01) and CG (*p* = 0.02), which did not continue at follow-up. There were no significant differences between EG and CG after the intervention or at follow-up. Strength: There were significant improvements in dynamic strength both after the intervention and at follow-up in EG (*p* < 0.001), for both paretic and non-paretic limbs. For CG, significant improvements in dynamic strength were found after intervention in the non-paretic limb (*p* < 0.05), but not in the paretic, and at follow-up only non-paretic flexion was significantly higher (*p* < 0.05) than at baseline. There were significant differences between EG and CG after the intervention (*p* < 0.001) and at follow-up (*p* < 0.001). There were significant improvements in isokinetic strength for both limbs both after the intervention and at follow-up in EG (*p* < 0.01). No differences were found for CG in isokinetic strength at intervention or follow-up. There was a significant difference between EG and CG (*p* < 0.05) after the intervention for non-paretic limb extension and flexion, and at follow-up for non-paretic limb extension. Gait: For EG, all gait tests improved significantly (*p* < 0.05) after the intervention, and that change was maintained at follow-up in TUG and 6MWT scores. For CG, only TUG improved significantly (*p* < 0.05) after the intervention. There were no significant differences between EG and CG between pre and post intervention, but there were significant differences at follow-up for TUG score (*p* < 0.05).
Mun et al. (2019) [36]	Two measurement time points (pre and post intervention) - Spasticity: Biodex system assessing the resistance to mobilization within different speed movements. - Gait: TUG - Balance: BBS; Platform for calculating load distribution in standing with eyes open and closed.	Spasticity: There was a decrease in spasticity in EG between pre and post intervention at angular velocities of 60°/sec, 180°/sec, and 240°/sec (*p* < 0.05). There was a decrease in spasticity in CG between pre and post intervention at angular velocities of 180°/sec and 240°/sec (*p* < 0.05). EG decreased spasticity statistically significantly compared to CG at angular velocities of 180°/sec (*p* = 0.02) and 240°/sec (*p* = 0.04). Gait: There was a significant decrease in the TUG score for both groups after the intervention (*p* < 0.05). No statistically significant differences were observed between the two groups (*p* = 0.11). Balance: There was a significant improvement in the BBS score for both groups after the intervention (*p* < 0.05). The BBS score in EG increased significantly compared to the CG (*p* < 0.01). An increase in weight distribution to the paretic side was found in both groups, both with eyes open and closed, after the intervention (*p* < 0.05). EG statistically significantly increased weight distribution with both eyes open (*p* = 0.04) and closed (*p* = 0.03) compared to CG.
Lattouf et al. (2021) [37]	Two measurement time points (pre and post intervention) - Spasticity: MAS - Gait: 10-m Walk Test; 6MWT - Strength: Maximum force calculated from Brzycki’s equation	Spasticity: There were no differences between or within the groups. Gait: For both groups, a significant difference was found in the time of the 10-m Walk Test (*p* ≤ 0.00001), with a higher walking speed after the intervention. No differences were found in the time of the 10-m Walk Test between the two groups after the intervention. A significant effect was observed in 6MWT between pre and post treatment for CG (*p* ≤ 0.0003) and for EG (*p* ≤ 0.0001). The results showed a statistically significant difference between the two groups (*p* ≤ 0.01).
		Strength: There was a significant difference between pre and post treatment for both groups (CG *p* ≤ 0.0001, EG *p* ≤ 0.0001). EG showed a significantly greater increase after treatment than CG (*p* ≤ 0.014).
Patten et al. (2013) [31]	Four measurement time points (pre-evaluation, in the rest period, at the end of the interventions, and at 6 months) - Spasticity: Ashworth Scale - Function: WMFT-FAS, UL-FMA, FIM	Spasticity: No significant changes were found in the Ashworth score at the post-intervention assessment or at 6 months (*p* > 0.05).
		Function: Significant improvements in WMFT-FAS were found after treatment block 1 in both groups (*p* < 0.05). These differences were significantly greater after the Hybrid Group compared to the Functional Physical Therapy group (*p* = 0.03). Tests of a period effect revealed greater improvements in WMFT-FAS after Hybrid versus Functional Physical Therapy (*p* = 0.02), regardless of where they occurred in the order of treatment. Overall, no differences were revealed because of the order of treatment (*p* = 0.43). A significant increase in WMFT-FAS was observed during the 6-month follow-up period (*p* = 0.03). No differences were revealed between Order A and Order B at the 6-month follow-up (*p* > 0.05). A significantly higher proportion of participants (51% vs. 39%) achieved the minimum significant difference of two points or more in the FIM after the Hybrid (*p* = 0.05). These positive changes were observed in 69% of participants at 6 months (*p* = 0.05). Post-intervention improvements were detected in both the total score and the shoulder-elbow portions of the UL-FMA, but these were not statistically significant. Significant differences were found for UL-FMA at 6 months after the intervention, with the minimum significant difference reaching 53% of all participants (*p* = 0.04)

6MWT, 6-Minute Walk Test; BBS, Berg Balance Scale; CG, Control Group; EG, Experimental Group; FIM, Functional Independence Measure; MAS, Modified Ashworth Scale; MMAS, Modified Modified Ashworth Scale; TUG, Timed Up & Go; UL-FMA, Upper Limb Fugl-Meyer Assessment; WMFT-FAS, Wolf Motor Function Test-Functional Abilities Scale.

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
