# Peer review of "Effects of Resistance Training on Spasticity in People with Stroke: A Systematic Review"

_brainsci, 2024, doi:10.3390/brainsci14010057_

Round 1

Reviewer 1 Report

Comments and Suggestions for Authors

Chacon-Barba et al are presenting an excellent systematic review on the effects of resistance training on post-stroke spasticity. The methods , results and discussion are very well presented. The only point I would like to mention is that the authors should include as limitation that the specific time of resistance training was not considered in the analysis.

Author Response

Reviewer 1

RV:

Chacon-Barba et al are presenting an excellent systematic review on the effects of resistance training on post-stroke spasticity. The methods, results and discussion are very well presented. The only point I would like to mention is that the authors should include as limitation that the specific time of resistance training was not considered in the analysis.

AA:

Thank you for reviewing our manuscript. We appreciate your valuable feedback and comments on our work. We have carefully considered your comments. Therefore, we added the following sentence into the last paragraph of the Discussion section:

In addition, the lack of information about the specific duration of resistance training in multimodal programs was not reported by some studies and therefore no solid conclusions could be drawn on this issue.”

Reviewer 2 Report

Comments and Suggestions for Authors

This study was a review of the effects of spasticity and resistance training in stroke patients. Please check the comments.

Abstract

Page. 1, line18 What is WOS?

Page. 1, line 21 Authors should describe the results in detail. How many hits did the paper find and how many were deleted?

Introduction

Page.1 line31- I thought that introduction needed a neurophysiological explanation for spasticity. Then please describe why resistance training is so effective.

Page 1, line 57 Do you conduct systematic reviews because the evidence is limited? I thought the purpose and background behind conducting this review was inadequate. Please clarify why this review is necessary.

Methodology

Page 3, line 90 What software did the author use to select the extracted papers? And how were they integrated? The procedure should be described in detail.

Additionally, is there a clear reason not to conduct a meta-analysis? I think I should add an analysis. This is because the author also extracts the effects.

Results

Page 4, line 120 In addition to placing the results on the flowchart, please add them to the Results section. The procedure, the number of exclusions, etc.

Table5. Different types of letters are used. Please correct.

Discussion

Page 12, line13 What can this Review be useful to the reader?

What do you think about the use of this evidence, including clinical?

Comments on the Quality of English Language

Nothing.

Author Response

Reviewer 2

RV:

This study was a review of the effects of spasticity and resistance training in stroke patients. Please check the comments.

Abstract

Page. 1, line18 What is WOS?

Page. 1, line 21 Authors should describe the results in detail. How many hits did the paper find and how many were deleted?

AA:

Thank you for reviewing our manuscript. We appreciate your valuable feedback and comments on our work. We have carefully considered your comments. Therefore, we added the meaning of WOS and included a more detailed description of the results in the Abstract section.

RV:

Introduction

Page.1 line31- I thought that introduction needed a neurophysiological explanation for spasticity. Then please describe why resistance training is so effective.

Page 1, line 57 Do you conduct systematic reviews because the evidence is limited? I thought the purpose and background behind conducting this review was inadequate. Please clarify why this review is necessary.

AA:

According to the reviewer’s suggestion, the Introduccion was modified to include the neurophysiological explanation for spasticity and the benefits of using resistance training as intervention. (Please refer to the Introduction section)

The last paragraph of the Introduction section was modified to better explain the rationale for conducting this systematic review. (Please refer to the Introduction section)

RV:

Methodology

Page 3, line 90 What software did the author use to select the extracted papers? And how were they integrated? The procedure should be described in detail.

Additionally, is there a clear reason not to conduct a meta-analysis? I think I should add an analysis. This is because the author also extracts the effects.

AA:

According to the reviewer’s comment, the following information was added to the subsection “2.1. Search strategy and selection process”: “The bibliographic information of the retrieved articles was imported into the Mendeley reference manager [25]. An initial manual check was performed to ensure accuracy, followed by grouping and sorting by title to eliminate duplicates. Titles and abstracts were then assessed, and those without human subjects and non-RCTs were discarded. Finally, compliance with inclusion criteria was annotated using the notes tool in Mendeley reference manager [25]. Articles that did not meet the established selection criteria were excluded by evaluating the full-text of the screened articles. The remaining studies were eligible for inclusion in the systematic review.”

A meta-analysis was not performed because of the high heterogeneity in terms of study interventions and outcome measures, as well as the different body regions assessed, so a meta-analysis is not congruent enough to extract a quantitative synthesis that adds qualitative value to the results of the studies analyzed. This information was added to the Limitations paragraph at the end of the Discussion section.

RV:

Results

Page 4, line 120 In addition to placing the results on the flowchart, please add them to the Results section. The procedure, the number of exclusions, etc.

AA:

A detailed information about the selection process was added, as follows:

A total of 274 articles were found in a first search and a total of 80 duplicate records were removed. The title and abstract of the remaining records (194) were screened and 171 were then excluded due to different reasons (not topic and not RCT). The full-text of the 23 remaining studies were assessed to verify the compliance of the eligibility criteria. Finally, nine articles were included in this review.”.

RV:

Table5. Different types of letters are used. Please correct.

AA:

It was corrected.

RV:

Discussion

Page 12, line13 What can this Review be useful to the reader?

What do you think about the use of this evidence, including clinical?

AA:

A final paragraph was added to the Discussion section detailing the scientific and clinical utility of this systematic review.

“Despite these limitations, the systematic review aims to contribute significantly to understanding the relationship between resistance training and spasticity post-stroke. The comprehensive evidence from this review enables clinicians to make informed decisions when considering the inclusion of resistance training in post-stroke rehabilitation plans. Recognizing its potential benefits in reducing spasticity, as well as improving critical functional areas such as strength, gait, and balance, enables the development of personalized and thorough treatment strategies. Furthermore, by highlighting existing gaps and limitations, this review acts as a catalyst for future research efforts. Finally, this systematic review offers clinicians valuable information for refining rehabilitation strategies, which may result in improved functional performance and quality of life for post-stroke, spastic patients.”

Reviewer 3 Report

Comments and Suggestions for Authors

This paper reports on a systematic review of the effects of resistance training (RT) on spasticity in people with stroke. The authors included 9 articles involving 225 participants. They found that RT was helpful in reducing spasticity as well as other benefits, but heterogeneity limited the validity of the findings.

There are some issues the authors may wish to attend to:

1.       Abstract – line 22 - to include data on sex and age of the analysed subjects

2.       Lines 31-32 – ‘heart attack’ is unlikely to be in the definition of stroke – please check

3.       Lines 34-35 - ‘In this sense’ can be deleted

4.       Line 48 – to add ‘with’ between ‘present’ and ‘spastic’

5.       Figure 1 – there is an error in the ‘n’ for ‘Reports sought for retrieval’

6.       Table 3 – please provide the units for age eg years

7.       Lines 212-214 – Modified Ashworth Scale appears twice…

8.       Line 273 – I believe the authors mean ‘rehabilitation’ and not ‘neurological’ trials (McIntyre A, Richardson M, Janzen S, Hussein N, Teasell R. The evolution of stroke rehabilitation randomized controlled trials. Int J Stroke. 2014 Aug;9(6):789-92)

Author Response

Reviewer 3

RV:

This paper reports on a systematic review of the effects of resistance training (RT) on spasticity in people with stroke. The authors included 9 articles involving 225 participants. They found that RT was helpful in reducing spasticity as well as other benefits, but heterogeneity limited the validity of the findings.

There are some issues the authors may wish to attend to:

1.       Abstract – line 22 - to include data on sex and age of the analysed subjects

AA:

Thank you for reviewing our manuscript. We appreciate your valuable feedback and comments on our work. We have carefully considered your comments. Therefore, we added data related with sex and age of the participants.

RV:

Lines 31-32 – ‘heart attack’ is unlikely to be in the definition of stroke – please check

AA:

It was replaced, as follows: “Stroke is defined as a sudden loss of neurological function resulting from an infarction or hemorrhage…”.

RV:

 Lines 34-35 - ‘In this sense’ can be deleted

AA:

It was deleted.

RV:

Line 48 – to add ‘with’ between ‘present’ and ‘spastic’

AA:

It was added.

RV:

Figure 1 – there is an error in the ‘n’ for ‘Reports sought for retrieval’

AA:

It was modified.

RV:

Table 3 – please provide the units for age eg years.

AA:

It was added.

RV:

Lines 212-214 – Modified Ashworth Scale appears twice…

AA:

The sentence was modified as follows: “Most studies employed the Modified Ashworth Scale [25,28–31,33]. The remaining studies used the Ashworth Scale [27], the Modified Modified Ashworth Scale [26], and the Biodex system [32],..”

RV:

Line 273 – I believe the authors mean ‘rehabilitation’ and not ‘neurological’ trials (McIntyre A, Richardson M, Janzen S, Hussein N, Teasell R. The evolution of stroke rehabilitation randomized controlled trials. Int J Stroke. 2014 Aug;9(6):789-92).

AA:

It was modified, as follows: “Concerning the characteristics of the studies included, samples were relatively small, involving an average of 30 subjects per study. In this sense, it is a common limitation in stroke rehabilitation trials, since it is difficult to obtain large sample sizes because patients are usually treated only in a neurological institution or center, costs are usually high, and inclusion criteria are very narrow [34,35].”.

A new reference was added:

McIntyre A, Richardson M, Janzen S, Hussein N, Teasell R. The evolution of stroke rehabilitation randomized controlled trials. Int J Stroke. 2014 Aug;9(6):789-92.

Round 2

Reviewer 2 Report

Comments and Suggestions for Authors

No comment.